# Effect of the Indentation Load on the Raman Spectra of the InP Crystal

**DOI:** 10.3390/ma15155098

**Published:** 2022-07-22

**Authors:** Dariusz Chrobak, Mateusz Dulski, Grzegorz Ziółkowski, Artur Chrobak

**Affiliations:** 1Institute of Materials Engenering, Faculty of Science and Technology, University of Silesia in Katowice, 75 Pułku Piechoty 1A, 41-500 Chorzow, Poland; mateusz.dulski@us.edu.pl; 2Institute of Physics, Faculty of Science and Technology, University of Silesia in Katowice, 75 Pułku Piechoty 1A, 41-500 Chorzów, Poland; grzegorz.ziolkowski@us.edu.pl (G.Z.); artur.chrobak@us.edu.pl (A.C.)

**Keywords:** semiconductors, InP, Raman spectroscopy, indentation

## Abstract

Nanoindentations and the Raman spectroscopy measurements were carried out on the (001) surface of undoped and S-doped InP crystal. The samples were indented with the maximum load ranging from 15 mN to 100 mN. The phase transition B3→B1 was not confirmed by spectroscopic experiments, indicating a plastic deformation mechanism governed by dislocations activity. Increasing the maximum indentation load shifts and the longitudinal and transverse optical Raman bands to lower frequencies reveals a reduction in the elastic energy stored in the plastic zone right below the indentation imprint. Mechanical experiments have shown that a shift in Raman bands occurs alongside the indentation size effect. Indeed, the hardness of undoped and S-doped InP crystal decreases as a function of the maximum indentation load.

## 1. Introduction

Indium phosphide is one of the technologically essential semiconductors for high-performance optoelectronic systems, such as sensors, optical integrated circuits, and laser diodes. Among others, the knowledge of InP mechanical properties is crucial to developing small-scale electronic devices [1]. In fact, it is of fundamental interest to investigate the mechanical properties of indium phosphide under conditions of contact pressures. A method of realizing this task combines nanoindentation testing with micro-Raman spectroscopy. This approach has been successfully used on many occasions, e.g., to identify a sequence of the indentation-induced phase transformations of Si [2,3,4] or to determine stress in silicon films on a sapphire substrate [5].

The Raman spectrum of the B3 phase (semiconducting zinc-blende lattice) of the InP crystal under normal conditions consists of the LO (longitudinal optical) and TO (transverse optical) bands located around $345\;{\mathrm{cm}}^{-1}$ and $303\;{\mathrm{cm}}^{-1}$, respectively, [6]. Lin et al. [7,8] showed that the sulfur doping shifts the LO and TO bands to $339\;{\mathrm{cm}}^{-1}$ and $302\;{\mathrm{cm}}^{-1}$, while, in the case of doping by iron, the positions of the main Raman bands are $339\;{\mathrm{cm}}^{-1}$ and $299\;{\mathrm{cm}}^{-1}$. Moreover, the presence of S or Fe atoms in the crystal lattice generates the local vibrational (LV) modes at $269\;{\mathrm{cm}}^{-1}$ and $418\;{\mathrm{cm}}^{-1}$, respectively. An increase in hydrostatic pressure causes the Raman bands’ blue shift, i.e., in the direction of a higher wavenumber. The LO and TO Raman peaks disappear after complete transformation into the B1 (metallic rock salt lattice) phase. Interestingly, doping by sulfur and iron decreases the pressure of the B3→B1 phase transition from 10.8 GPa (undoped InP) to 10.4 GPa and 8.2 GPa, respectively. In contrast, the doping causes an increase in microhardness [9] and yield point [10,11] while preventing the formation of dislocations during crystal growth [12].

The Raman spectroscopy method was also used to study the phase composition of a plastically deformed zone right under the residual indentation imprint. The goal was to answer the question of which phenomenon, phase transformation or nucleation of dislocations starts the plastic deformation in an initially dislocation-free crystal volume. The available literature indicates dislocation generation as a cause of nanoindentation-induced plasticity in InP crystal [13,14,15,16], without revealing the transformation from initial B3 to the high-pressure B1 phase. Raman spectroscopy measurements performed at the center of a residual impression made by nanoindentation are an example of such investigation [17].

Raman microscopy is often used as an auxiliary method in characterizing the crystal response to the applied contact load [17,18,19,20,21]. Hence, the motivation of the present research was to more systematically approach the Raman spectroscopic characterization of the small-scale plasticity of InP crystal. This paper presents the results of high-load nanoindentation experiments combined with Raman spectroscopy measurements performed on undoped and S-doped InP crystals. We studied both the effect of doping and the effect of residual stresses on the position of Raman bands. The analysis of nanoindentation experiments showed a relationship between the Raman bands’ shift and the indentation size effect [22].

## 2. Materials and Methods

Load-controlled nanoindentations (Triboindenter TI-950, Berkovich diamond probe) were carried out on the (001) surface of the undoped and S-doped (carrier concentration 1.7–1.9 × 10^18^ cm^−3^) InP wafers fabricated by the vertical gradient freeze method. The undeformed substrates were spectroscopically characterized. We obtained the following results: $\omega_{LO}=346.6\pm 0.5\;{\mathrm{cm}}^{-1}$, $\omega_{TO}=306.9\pm 0.3\;{\mathrm{cm}}^{-1}$ and $\omega_{LO}=346.9\pm 0.4\;{\mathrm{cm}}^{-1}$, $\omega_{TO}=305.8\pm 0.5\;{\mathrm{cm}}^{-1}$ for the undoped and S-doped substrate, respectively. Using pristine substrates, we performed 20 nanoindentations per each maximum load: 15, 25, 35, 55, 75, and 100 mN. The applied load functions ensured the same loading rate of 10 mN/s for all mechanical experiments. The topography of residual impressions was recorded using scanning probe microscopy (SPM)—one of the triboindenter’s capabilities.

The analysis of the nanoindentation experiments was performed using the Oliver–Pharr method [23]. The contact depth $h_{c}$, the hardness *H*, and the reduced Young’s modulus $E_{r}$ were calculated for each load–displacement P(h) curve. The contact depth depends on the maximum indentation depth $h_{max}$, the maximum indentation load $P_{max}$ and the contact stiffness *S*:(1)$$h_{c}=h_{max}-\varepsilon \frac{P_{max}}{S},\;S=\frac{dP}{dh}\left(h_{max}\right)$$
where ε=0.75. The calculation of the hardness and the reduced Young’s modulus requires the determination of the contact area *A* as a function of the contact depth $h_{c}$:(2)$$A=A\left(h_{c}\right),\;H=\frac{P_{max}}{A},\;E_{r}=\frac{\sqrt{\pi }}{2}S\frac{1}{\sqrt{A}}$$

In order to investigate the shape of the contact area function $A(h_{c})$, we performed 50 indentations on the fused-quartz sample and came to the conclusion that the following formula:(3)$$A\left(h_{c}\right)=c_{0}h_{c}^{2}+c_{1}h_{c},\;c_{0}=19.43,\;c_{1}=3322.48$$
gives acceptable results. Indeed, by applying the above contact area function to the fused quartz, we obtained the hardness H=9.29±0.17 and the reduced Young’s modulus $E_{r}=68.2\pm 1.0$ GPa, which are close to the reference data: H=9.25±0.93 and $E_{r}=69.6\pm 3.4$ GPa. Equation (Equation 3) is composed of the first two terms of the general formula for the contact area function used in the Oliver–Pharr method [23]. The fused (amorphous) quartz, due to its isotropic mechanical properties, is a widely used reference material for the calibration of nanoindentation systems. For this reason, smaller errors in the hardness measurements and Young’s modulus can also be expected [23].

The Raman experiment was performed using WITec confocal Raman microscope CRM alpha 300R equipped with an air-cooled solid-state laser (λ=532nm, P=2mW) and a CCD detector. The excitation laser radiation was coupled with a microscope through a single-mode optical fiber. The 100×/0.90 NA Olympus MPLAN air objective was chosen. The diameter of the pinhole equal to 25 µm was chosen as an optimal solution to preserve the compromise between lateral and depth resolution and ensure to avoid a loss in the z direction. According to this, the lateral resolution (LR) was estimated using the Rayleigh criterion LR=0.61λ/(NA), while a depth resolution (DR) as $DR=\lambda /{\left(NA\right)}^{2}$, where LR is the minimum distance between resolvable points (in X, Y direction), DR is the minimum distance between resolvable points (in the z direction), NA is the numerical aperture, and λ is the wavelength of laser excitation. As a result, LR=0.36 μm while DR=0.65 μm. Raman scattered light was focused on a multi-mode fiber and then on a spectral monochromator previously calibrated using the emission lines of a Ne lamp. Instrument calibration was verified by checking the position of the Si ($520.7\;{\mathrm{cm}}^{-1}$). Surface Raman imaging map was collected in a 15 μm × 15 μm area using 60×60 pixels (3600 spectra) with an integration time of 0.5 s per spectrum, and a precision of moving the sample during the measurements with a precision of ±0.5 μm. The total exposure time for a map was estimated at ca. 30 min. The spectral resolution of $1\;{\mathrm{cm}}^{-1}$ was achieved using an 1800 line/mm grating. The output data were manipulated by performing a baseline correction using the auto-polynomial function of degree 3 and were submitted to a cosmic ray removal procedure. Chemical images were generated using a sum filter that integrating the intensity over a defined frequency range (e.g., LV, LO or TO modes). This procedure was applied to preliminarily recognize the structural differentiation of the analyzed sample. Then, a more detailed analysis was performed using cluster analysis (CA) to group the objects (spectra from the map) into clusters. K-means analysis with the Manhattan distance for all Raman imaging maps was carried out to distinguish how structurally affected spectra resulted from the indentation and find the Raman spectrum of the most central part of the indent (Appendix A). This analysis was performed using WITec ProjectFive Plus software. Finally, a bands’ fitting analysis using a Lorentz–Gauss function with the minimum number of the component was performed using the GRAMS 9.2 spectroscopy software suite (Thermo Scientific, Walthamm, MA, USA).

## 3. Results and Discussion

Figure 1 shows exemplary Raman spectra recorded at the center of residual impression ($P_{max}=25\;\mathrm{mN}$, $P_{max}=100\;\mathrm{mN}$) made on the (001) surface of the S-doped InP crystal. For the highest maximum indentation load $P_{max}=100\;\mathrm{mN}$, the most intense peaks correspond to the longitudinal optical (LO) and transverse optical (TO) modes at $\omega_{LO}=344.3\;{\mathrm{cm}}^{-1}$ and $\omega_{TO}=303.2\;{\mathrm{cm}}^{-1}$ (see also Appendix A). A similar Raman spectrum was obtained for the surface imprint performed with $P_{max}=25\;\mathrm{mN}$: $\omega_{LO}=345.4\;{\mathrm{cm}}^{-1}$ and $\omega_{TO}=305.0\;{\mathrm{cm}}^{-1}$. One can see that the Raman spectrum is affected by incorporating the sulfur atoms into the B3 phase lattice of InP. As a consequence, the LV band arises at a low frequency of $\omega_{\mathrm{LV}}=250.5\;{\mathrm{cm}}^{-1}$ for $P_{max}=100\;\mathrm{mN}$ and $\omega_{\mathrm{LV}}=252.8\;{\mathrm{cm}}^{-1}$ for $P_{max}=25\;\mathrm{mN}$.

An influence of the maximum indenter load $P_{max}$ as well as sulfur doping on the position of the Raman bands is shown in Figure 2. The Raman mode wavenumber corresponding to each $P_{max}$ value is the arithmetic mean of five measurements—errors do not exceed $0.6\;{\mathrm{cm}}^{-1}$. The S-doping of InP crystal shifts the LO and TO bands in the direction of the minor wavenumbers, which is a consequence of the fact that doping by sulfur slightly increases the lattice constant of the B3 phase from 5.8610 Å to 5.8626(1) Å [7]. Indeed, according to the work by Angel et al. [24], the relationship between the Raman mode shift $\Delta \omega_{mode}$ and the change of the unit cell volume ΔV (or lattice constant) can be expressed by the following equation, valid for cubic crystals:(4)$$\frac{-\Delta \omega^{mode}}{\omega_{0}^{mode}}=\gamma_{1}^{m}\frac{\Delta V}{V_{0}}$$
where: $V_{0}$—the volume of strain-free unit cell; $\omega_{0}^{m}$—frequency of the Raman mode of the strain-free crystal; $\gamma_{1}^{m}$—the Raman mode Grüneisen factor (for undoped InP: $\gamma_{1}^{LO}=1.24$, $\gamma_{1}^{TO}=1.44$ [6]). Thus, the expansion of the InP crystal lattice, caused by doping, results in the experimental observation of a negative change in band wavenumber.

The Raman band wavenumber also depends on the maximum indentation load $P_{max}$. It was found (Figure 2) that the LO, TO, and LV phonon modes shift approximately linearly into lower frequencies in response to the increasing value of the maximum indentation load:(5)$$\begin{array}{cc} \omega_{LO,undoped} & =347.3-0.0289P_{max}\\ \omega_{TO,undoped} & =306.9-0.0223P_{max}\\ \omega_{LO,doped} & =346.4-0.0224P_{max}\\ \omega_{TO,doped} & =306.1-0.0325P_{max}\\ \omega_{LV,doped} & =253.9-0.0348P_{max} \end{array}$$

An inverse phenomenon was observed for silicon [25]. The measurements performed for the pristine crystal surface showed the major Raman band distributed around $520\;{\mathrm{cm}}^{-1}$, whereas the measurements performed at the center of the residual imprint showed the band shifted up by ∼5$\;{\mathrm{cm}}^{-1}$ (so-called blue shift). The authors argued that the effect is caused by the presence of residual stresses (dislocations) in the post-indentation plastic zone. Thus, the red shift of the LO, TO, and LV Raman bands observed for our InP crystals should be related to a reduction in the residual stresses in the plastic zone. Because the level of the residual stresses coincides with the density of dislocations, our results (Figure 2) obtained for InP can be interpreted as a consequence of the dislocation density decrease.

Looking for additional arguments that can support our conclusion derived from the Raman spectroscopy measurements, we turned our attention to nanoindentation experiments. In particular, we investigated how the hardness *H* and the reduced Young’s modulus $E_{r}$ depend on the indentation depth $h_{c}$ (or equivalently the maximum indentation load $P_{max}$). In order to calculate *H* and $E_{r}$ we analyzed the P(h) curves (Figure 3a) using a common approach proposed by Oliver and Pharr (see Methods).

The comparison of the unloading segments of the P(h) curves (Figure 3a) shows that S-doping reduces the depth of the residual impression and, simultaneously, the value of the contact depth $h_{c}$ (Table 1). Our calculation also showed that the mean value of $E_{r}$ (Figure 3b, right scale) is almost constant with respect the maximum indentation load. However, $E_{r}$ decreases from 96.5±3.3 GPa to 88.6±6.4 GPa in response to S-doping. In contrast, the hardness *H* of the S-doped crystal is greater than the hardness measured for the undoped sample in the applied maximum load range. Crucial information is contained in the dependence of hardness *H* on the contact depth $h_{c}$ (or equivalently $P_{max}$). Figure 3b shows a characteristic decrease in the hardness observed for undoped and S-doped crystals, indicating the so-called indentation size effect (ISE) [22]. According to the general theory of ISE, the relationship between the hardness and the contact depth can be described by the following formula:(6)$$H=H_{0}\sqrt{1+\frac{h^{*}}{h_{c}}}$$
where $H_{0}$ is the macroscopic hardness and $h^{*}$ is the length parameter. Fitting the above equation to the experimental data gives: $H_{0}=4.92\;\mathrm{GPa}$, $h^{*}=114.8\;\mathrm{nm}$ for undoped and $H_{0}=5.08\;\mathrm{GPa}$, $h^{*}=132.6\;\mathrm{nm}$ for S-doped InP crystal. The ISE is caused by the decreased density of geometrically necessary dislocations (GNDs). Although they are only a part of all dislocations generated during nanoindentation, the total dislocations density may decrease in some range of the contact depths represented by the $h^{*}$ parameter.

The dislocation density reduction in a post-indentation plastic zone should also be reflected by lowering the density *w* of the plastic work performed during the indentation cycle. Indeed, this quantity refers to the amount of work per volume unit used for the generation of the dislocation structure:(7)$$w=\frac{W_{l}-W_{u}}{V}$$
where: $W_{l/u}$ is the work performed during loading/unloading; and *V*—the post-indentation plastic zone volume. To see how *w* depends on $P_{max}$, we integrated each P(h) curve to obtain loading $W_{l}$ and unloading work $W_{u}$. Then, to estimate the volume of the plastic zone, the cavity model developed by Johnson [26] was used (Figure 4a). The calculations performed for each P(h) curve were executed in three steps:In the work by Yang et al. [27], one can find the following equation:
(8)$$\frac{H}{E_{r}}=\frac{2\left(1-\nu \right)\cot\alpha }{3+1/\ln\left(c/a\right)}\frac{W_{u}}{W_{l}}$$
where: *a*—the radius of the contact area between the indenter and the crystal surface; $\alpha =70.32^{\circ }$—the effective semi-angle of the conical tip equivalent to the Berkovich indenter; ν = 0.35—the Poisson ratio of InP. We applied the Equation (Equation 8) to obtain c/a—the ratio of the plastic zone radius *c* to the contact radius *a*.In order to calculate the contact radius *a*, we expressed the contact area in two equivalent forms: as a function of the contact depth: $A=c_{0}h_{c}^{2}+c_{1}h_{c}$ and the contact radius: $A=\pi a^{2}$. Hence, $a=\sqrt{\frac{c_{0}h_{c}^{2}+c_{1}h_{c}}{\pi }}$.We used the $V=2/3\pi c^{3}$ equation that gives the required approximation of the post-indentation plastic zone volume (half of sphere).Calculated densities of the plastic work *w* were averaged for each value of the maximum indentation load.

The results of calculations are presented in Figure 4b and Table 2. It can be seen that the $H/E_{r}:W_{u}/W_{l}$ ratio slightly decreases within the range of $P_{max}$ and, consequently, it entails a small reduction in c/a ratios. Moreover, the plastic zone volume calculated for both undoped and S-doped InP are almost equal within an accuracy given by error ranges. Finally, the density of the plastic work is a decreasing function of the maximum indentation load which confirms the reduction in the density of dislocations generated during the full indentation cycle.

In order to explain our choice of the indirect method for estimation *c*, we refer to the work by Yan et al. [28]. The electron microscopy observations of indentation-induced plastic zones made on (001) surface of InP (maximum indentation load of 10 mN and 50 mN) show the highly irregular shape of their boundaries. Only one thing can be said: the area with high dislocation density is larger in the case of $P_{max}=100\;\mathrm{mN}$. Thus, the experimental estimation of the radius *c* is highly problematic.

## 4. Conclusions

In conclusion, nanoindentation experiments with the maximum indentation load ranging from 15 mN to 100 mN were performed on the (001) surface of undoped and S-doped InP crystal. The mechanical tests were accompanied by the Raman spectroscopy measurements performed at the center of the residual indentation imprints. An increase in the maximum indentation load caused the linear shift of the LO, TO, and LV Raman bands in the direction of the minor wavenumbers (red shift). This phenomenon indicates a reduction in the post-indentation (residual) stresses in response to the increasing maximum indentation load.

An analysis of the nanoindentation P(h) curves shows that the hardness decreases as a function of the contact depth, indicating the participation of the indentation size effect. The hardness of S-doped InP is greater than that of the undoped crystal. In contrast, the reduced Young’s modulus is less. The reduction in the dislocation density predicted by the Raman spectroscopy measurements was also confirmed by calculating the density of the plastic work performed during the complete indentation cycle.

## Figures and Tables

**Figure 1 materials-15-05098-f001:**
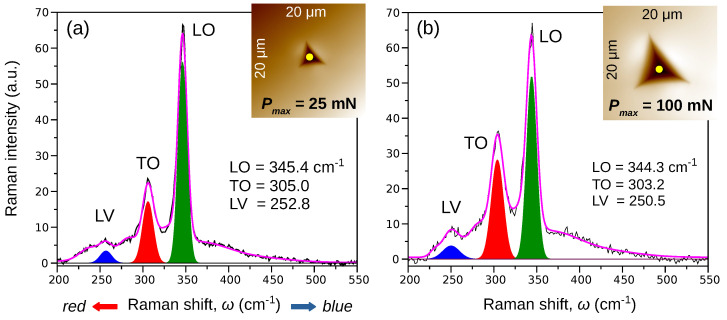
Results of the Raman spectroscopy measurements made on the (001) surface of the S-doped InP crystal. Bands’ locations were measured at the center of the residual indentation imprint with the maximum load (**a**) $P_{max}=25\;\mathrm{mN}$ and (**b**) $P_{max}=100\;\mathrm{mN}$. The Lorentz–Gauss function was used for spectral fitting to expose the main modes: LO, TO, and LV. The magenta curves represent results of fitting to the raw data (black curves). Insets present the SPM images of the post-indentation imprints. The yellow circles represent laser spots focused on the bottom of the imprints.

**Figure 2 materials-15-05098-f002:**
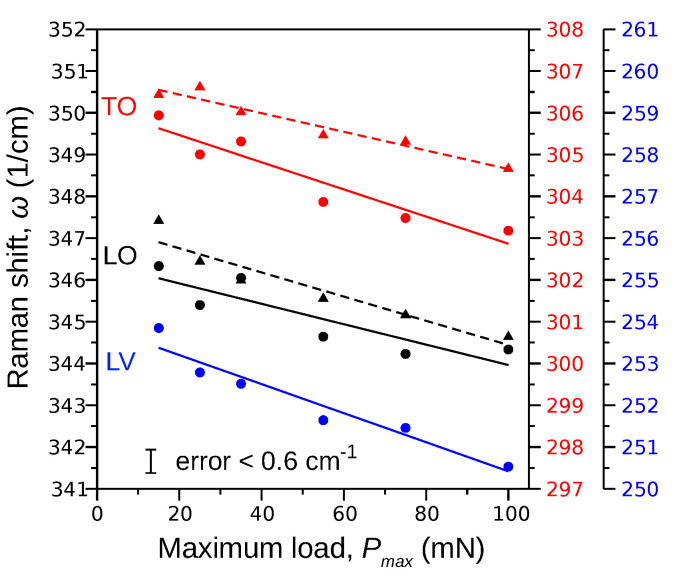
Effect of the maximum indentation load $P_{max}$ on the LO, TO and LV modes of the undoped (triangles, dashed lines) and S-doped (circles, solid lines) InP crystal. The doping as well as an increase in $P_{max}$ shift the Raman bands in the direction of lower wavenumbers (red shift).

**Figure 3 materials-15-05098-f003:**
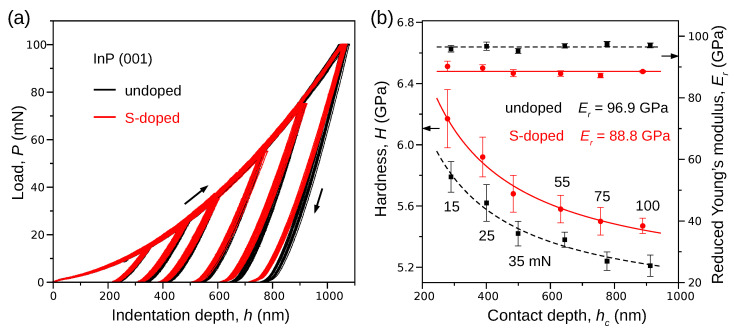
Results of the nanoindentation experiments performed on the (001) surface of undoped (red) and S-doped (black) InP crystal. The left black arrow indicates the direction of loading, whereas the right black arrow shows the direction of unloading. (**a**) The unloading segments of the P(h) curves show a decrease in the depth of the indentation imprint due to doping. (**b**) S-doping of InP crystal decreases $E_{r}$ simultaneously decreasing the values of *H*. The characteristic dependence of hardness on the contact depth (or equivalently, the maximum indentation load $P_{max}$) is a sign of the indentation size effect.

**Figure 4 materials-15-05098-f004:**
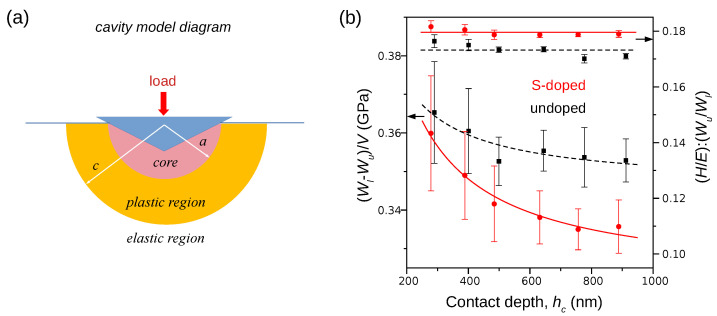
Results of the nanoindentation experiments analysis: (**a**) schematic of Johnson’s cavity model; (**b**) the plastic work density $(W_{l}-W_{u})/V$ decreases as a function of indentation contact depth $h_{c}$.

**Table 1 materials-15-05098-t001:** Results of the mechanical analysis of the nanoindentation experiments on undoped and S-doped InP crystal ($P_{max}$—the maximum load in mN, $h_{c}$—the contact depth in nm, *H*—the hardness in GPa, $E_{r}$—the reduced Young’s modulus in GPa).

	Undoped			S-Doped		
$P_{max}$	$h_{c}$	*H*	$E_{r}$	$h_{c}$	*H*	$E_{r}$
15	289.5±2.9	5.8±0.1	95.9±1.1	278.6±5.4	6.2±0.2	91.7±1.7
25	400.7±5.0	5.6±0.1	96.7±1.4	388.5±4.8	5.9±0.1	89.6±1.1
35	499.2±3.9	5.4±0.1	95.2±0.8	484.0±6.1	5.7±0.1	88.0±1.1
55	644.7±3.3	5.4±0.1	96.9±0.7	632.0±5.7	5.6±0.1	87.9±0.9
75	776.9±5.1	5.2±0.1	97.4±0.8	756.6±6.6	5.5±0.1	87.2±0.7
100	911.9±6.4	5.2±0.1	99.1±0.8	888.4±4.0	5.5±0.1	88.5±0.3

**Table 2 materials-15-05098-t002:** Geometrical characteristics of the plastic zone around the indentation imprint as a function of the maximum indentation load $P_{max}$ (mN): *a*—the contact radius (nm); *c*—the plastic zone radius (nm); *V*—the volume of plastic zone (μm^3^).

	Undoped			S-Doped		
$P_{max}$	*a*	*c*	*V*	*a*	*c*	*V*
15	908±07	1237±10	4.0±0.1	880±14	1221±09	3.8±0.1
25	1190±13	1614±11	8.8±0.2	1160±12	1602±11	8.6±0.2
35	1439±10	1939±07	15.3±0.2	1400±15	1922±11	14.9±0.3
55	1804±08	2433±07	30.2±0.3	1772±14	2432±08	30.1±0.3
75	2134±13	2848±13	48.4±0.7	2084±17	2860±08	49.0±0.4
100	2472±16	3286±10	74.3±0.7	2413±10	3287±13	74.4±0.9

## Data Availability

Not applicable.

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
