# Peer review of "Effect of the Indentation Load on the Raman Spectra of the InP Crystal"

_materials, 2022, doi:10.3390/ma15155098_

Round 1
Reviewer 1 Report
This research work has scientifically observed the effect of pressure on InP crystals, and provided a convincing explanation for the results, but there are still some small problems in the article. Therefore, I recommend that this work can be considered to publish in Materials after minor revision.
â‘ Lack of characterization of the crystal properties before applying pressure, only wavenumbers corresponding to some peaks can be seen in Figure 2.
â‘¡The formula for contact area suddenly uses fused-quartz samples, please specify the reason and explain the generality of the formula (method).
â‘¢It will be better if the parameters such as laser power, exposure time and number of cycles for refined Raman characterization were manifested.
â‘£It could be better if a Raman mapping image was added. For details, please refer to the paper “Electrochemically Exfoliating MoS2 into Atomically Thin Planar-Stacking Through a Selective Lateral Reaction Pathway”
Author Response
-

Reviewer 2 Report
Using Raman spectroscopy combined with nanoindentation experiment, it was confirmed that the plastic deformation mechanism on InP crystal is governed by dislocations more than by a phase transition mechanism. This paper is well written and correctly documented. May be it could also have been better positioned in relation to the other works of these authors.
In line 71, the supplementary material is not referenced and not available
Author Response
-
